# Targeted NGS in Diagnostics of Genodermatosis Characterized by the Epidermolysis Bullosa Symptom Complex in 268 Russian Children

**DOI:** 10.3390/ijms232214343

**Published:** 2022-11-18

**Authors:** Kirill Savostyanov, Nikolay Murashkin, Alexander Pushkov, Ilya Zhanin, Elkhan Suleymanov, Mariya Akhkiamova, Olga Shchagina, Elena Balanovska, Roman Epishev, Aleksander Polyakov, Andrey Fisenko

**Affiliations:** 1FSAI «National Medical Research Center for Children’s Health» of the Russian Federation Ministry of Health, 119991 Moscow, Russia; 2Ministry of Public Health, Republic of Chechnya, 364037 Grozny, Russia; 3Research Centre for Medical Genetics, 115522 Moscow, Russia

**Keywords:** epidermolysis bullosa, children, next generation sequencing, founder effect, microsatellite markers

## Abstract

The pathogenic variants of genes encoding proteins, participating in the formation and functioning of epidermis and dermo-epidermal junctions, create a large variety of clinical phenotypes from: small localized to severe generalized dermatitis, as well as early, or even, prenatal death due to extensive epidermis loss. The diagnostic panel in this study was developed for the purposes of identifying these pathogenic genetic variants in 268 Russian children, who possessed the epidermolysis bullosa symptom complex in a selection of 247 families. This panel included the targeted areas of 33 genes, which are genetic variants that can lead to the development of the phenotype mentioned above. The usage of next generation sequencing allowed the revelation of 192 various altered alleles (of which 109 alleles were novel, i.e., had not been described previously). In addition, it allowed the definition of the genetic variants that are both typical for most of the examined children and for the separate ethnic groups inhabiting modern Russia. We found that the most characteristic mutations for the Dargin and Chechen ethnic groups are the c.3577del deletion in the *COL7A1* gene and the c.2488G>A missense mutation in the *COL17A1* gene, respectively. In addition, the study of haplotypes of microsatellite markers, which we managed to conduct in the Dargin population, confirmed the presence of the founder effect.

## 1. Introduction

Epidermolysis bullosa (EB) is a group of disorders characterized by severe skin lesions in the form of blisters, mucous membrane injuries, as well as nails, hair, and teeth defects. The muscles, heart, bones, digestive tract, kidneys, and urinogenital tract can all be involved in the pathological process, thereby affecting the integrity and mechanical stability of the skin forming complexes via desmosomes, hemidesmosomes, basal membranes, and anchoring fibrils [1,2]. Meanwhile, less severe manifestations of skin fragility can be connected to such diseases as ichthyosiform erythroderma, superficial epidermolytic ichthyosis, pachyonychia congenita, porphyria cutanea, acrodermatitis enteropathica, etc. [3].

Within the last three decades, the mapping of new genes responsible for hereditary skin diseases has been based on the findings found in association and linkage studies of genetic markers or chromosomal areas with certain phenotypical signs. The usage of this slow, difficult, and—usually—unsuccessful approach has led to the identification of several new genes associated with skin fragility, due to efforts of molecular biologists and geneticists at the end of the 20th century [4,5]. Methods of genomic DNA amplification, in turn, have simplified the detection of mutations in these genes. This has led to an improvement in diagnosis specification, genetic counselling, and prenatal diagnostics. Further, it has also created the basis for a development of new biological therapy methods [6]. Previously, stepwise Sanger sequencing of the genes responsible for the development of the disease was used for the purposes of genetic diagnoses of hereditary diseases, united by a similar symptom complex [7]. However, in the last ten years, the process of “hunting for genes” and the identification of pathogenic variants have been optimized with the advent of the new era of next generation sequencing (NGS). This methodology has changed not only the strategies for finding genetic causes in clinical practice [8], but also the technology for fundamental discoveries in the field of hereditary skin diseases [9].

Prior to the introduction of NGS in the process of the diagnostics of hereditary dermatosis, which is characterized by the epidermolysis bullosa symptom complex, a very difficult and long diagnostic algorithm was used. This algorithm included immunofluorescence (IF) and/or microscopic analysis of the material collected from an invasive procedure (e.g., a skin biopsy) for the purposes of determining which genes’ coding nucleotide sequences could undergo bidirectional Sanger sequencing. In most cases, this choice could not be limited to one gene because immunofluorescence mapping/transmission electron microscopy often allowed one to only define major types of EB [3].

To date, there are four known major types of EB [10,11]. EB simplex is caused by the mutations within 14 different genes: *TGM5* [12], *PKP1* [13], *DSP* [14], *JUP* [15], *CHST8* [16], *CDSN* [17], *KRT5*/*KRT14* [18], *DST* (*BPAG1*) [19], *EXPH5* [20], *ITGA6* [21], *ITGB4* [22], *PLEC* [23], and *KLHL24* [24]. Junctional EB is caused by mutations in 8 different genes: *LAMA3*/*LAMB3*/*LAMC2* [25], *ITGA6*/*ITGB4* [26], *ITGA3* [27], *COL17A1* [28], and *CD151* [29]. Dystrophic EB is caused by mutations in the *COL7A1* gene [30]. Kindler syndrome is caused by mutations in the *FERMT1* gene [31]. Moreover, mutations in the *KRT1* [32] and *KRT10* [33] genes led to the development of congenital ichthyosiform erythroderma (CIE); mutations in the *KRT2* gene [34]—superficial epidermolytic ichthyosis (SEI); mutations in the *KRT6A*, *KRT6B*, *KRT16*, and *KRT17* genes—pachyonychia congenita; mutations in the *UROD* and *UROS* genes—porphyria cutanea and porphyria erythropoietic; mutations in the *SLC39A4* gene—acrodermatitis enteropathica. Mutations in all mentioned genes can also lead to the development of blisters on the skin; thus, we have decided to include them in one panel consisting of the targeted areas of the 33 genes and have developed them for the purposes of revealing the molecular basis of diseases that are within the epidermolysis bullosa symptom complex. A similar molecular diagnostic algorithm has been used prior for the purposes of genodermatosis diagnostics [35,36], and even for EB diagnostics [37,38]. However, this panel, which consists of coding and intronic areas of 33 genes, was used for the first time for the Russian population.

## 2. Results and Discussion

High efficiency and simultaneous sequencing of multiple genes typical for NGS allowed us to perform molecular genetic testing and to identify the genetic causes of hereditary dermatosis with the EB symptom complex in each of 268 (100%) children within 2–4 weeks from the moment the biomaterial was delivered to the laboratory. The most common form of epidermolysis bullosa was dystrophic EB; it was detected in 178 children from 166 (67%) families. The second most common form was EB simplex; it was detected in 60 children from 52 (21%) families. Junctional EB was detected in 18 (7%) families. Congenital ichthyosiform erythroderma and Kindler syndrome were detected in 7 (3%) and 3 (1.2%) families, respectively. In addition, superficial epidermolytic ichthyosis was confirmed in a single patient (Figure 1).

Nowadays, there is no published scientific research that covers the multiplex molecular diagnostics of EB in children using the NGS method. However, there are two recent studies describing small cohorts of patients with EB from various age groups. The first one has shown the data similar to our research—i.e., a considerable prevalence of the dystrophic EB form [37]—whereas the other study has shown an insignificant dominance of the EB simplex form [38]. It is considered that EB simplex is the most common type of EB; however, there are data contradicting this judgment [39]. Most likely all the studies using Sanger sequencing did not conduct a simultaneous research of several genes and always stopped after detecting pathogenic variants in one gene only. NGS is advantageous as it allows us to explore the targeted areas of all genes, as well as mutations that can lead to EB development, both simultaneously within one diagnostic pool. This can change the prevalence of various EB types due to the increase in the number of studies, as well as the size of patient cohorts.

The analysis of molecular genetic characteristics of the described diseases’ development has allowed us to estimate the mutability of 11 genes in examined children via the analysis of distribution of 192 various genetic variants. (Figure 2). It is worth mentioning that rare and novel genetic variants were also detected in other examined genes, but molecular genetic testing of parents did not allow us to assume these variants to be causative. We assumed that variants were not causative, because they were inherited from parents without clinical manifestations of EB (Table A1). However, they may still have had a modifying role [40].

Mutations in the *COL7A1* gene (NM_000094.4) were the most frequent and variable, causing dystrophic EB in 178 children from 166 (67%) families (Figure 2). Among 130 (68%) various mutations, 66 (53%) were novel, i.e., not described in the HGMD database. The mutation spectrum included 55 missense mutations, 18 nonsense mutations, 17 frameshift deletions, 11 frameshift duplications, 26 splicing mutations, one regulatory mutation (located in the promoter region), one in-frame deletion, and one mutation affecting the stop codon and subsequently the translation termination.

Three mutations among the other most common mutations in the *COL7A1* gene can be considered to be specific for Russian children with autosomal-recessive dystrophic EB. These mutations are: missense mutation c.425A>G, which was detected on 34 (9.5%) alleles in 27 (16%) non-related families and described previously as the most common mutation in the *COL7A1* gene [41]; new deletion c.3577del detected 20 times in 9 (5%) non-related families; and splicing mutation c.682+1G>A as described previously [42] and detected 19 times in 17 (10%) non-related families. We should note that this deletion: c.3577del, p.(A1193Lfs*72) was detected in a homozygous state in 9 non-related children living in Dagestan, 7 of whom are of Dargin origin.

Among other variants, a non-described duplication c.497dup, p.(V168Gfs*12) occurred on 11 (3%) alleles in 11 unrelated families. Further, duplication c.5261dup—p.(G1755Rfs*17), missense mutation c.5499C>T leading to a synonymous substitution p.(Gly1833=) and missense mutation c.6205C>T—p.(Arg2069Cys) were met on 10 alleles.

The most frequent mutation causing the development of autosomal-dominant dystrophic EB was the c.6127G>A, p.(G2043R) missense mutation. It was detected in 10 (4%) families from different regions of our country. This mutation was previously described as a typical mutation for the dominant form of dystrophic EB [43,44]. It has been established that the change in a glycine residue to any other amino acid residue leads to the most serious functional disorders in encoded collagen proteins [45]. Such mutations located in the *COL7A1* gene central domain are capable of causing autosomal-dominant dystrophic EB, due to the fact that the damage on one chromosome is enough for disease progression. In our research, we established that apart from the c.6127G>A, p.(G2043R) mutation (mentioned above), two other mutations (i.e., c.6191G>T, p.(G2064V) and c.6218G>A, p.(G2073D))—located in neighboring exons of the central domain—also lead to the development of the dominant form of dystrophic EB. Aside from that, a heterozygous mutation c.8371C>T, p.(R2791W) located in the C-terminus of the *COL7A1* gene can cause a mild phenotype of dominant dystrophic EB. In addition, because of this fact, a brother and a sister from one family were falsely diagnosed with EB simplex prior to molecular genetic testing. It is worth noting that the same mutations in the *COL7A1* gene can also lead to various phenotypic manifestations of the disease, even within one family. Thereby, prognosis on the disease development, considering the detected genotype, can be very difficult. Moreover, it was also confirmed by the results of foreign studies [46]. Other mutations were detected less than ten times each (Figure 3).

Moreover, 81 (62%) variants of the *COL7A1* gene were revealed only once. Almost all missense variants’ pathogenicity was confirmed by four bioinformatic resources, except for two variants (c.403C>A and c.2366G>T). Their pathogenicity was confirmed by three resources.

The variety of the detected mutations, including pathogenic variants that have not been described previously, can represent the considerable polymorphism of the *COL7A1* gene among different ethnic groups inhabiting modern Russia. It is necessary to note that the c.425A>G mutation is the most frequently encountered mutation, and this is not only the case in the context of the *COL7A1* gene. As such, it was detected in every tenth studied Russian family with EB.

The c.3577del, p.(A1193Lfs*72) deletion was detected in a homozygous state in 10 patients from 9 non-related families from the Republic of Dagestan. Dagestan is the most nationally diverse region of Russia: the people of Dagestan speak languages from four linguistic groups. A total of 14 native ethnic groups are officially established; aside from these, 14 minor ethnic groups can also be distinguished [http://www.demoscope.ru/weekly/ssp/rus_nac_cen.php?reg=6, accessed on 16 November 2022]. The people of Dagestan are characterized by a distinct ethnic and territorial assortativity. Seven families with the c.3577del mutation considered themselves to be of Dargin origin. According to the 2010 Russian population census [https://gks.ru/free_doc/new_site/perepis2010/croc/perepis_itogi1612.htm, accessed on 16 November 2022], 16.85% of all Dagestan residents are Dargins. The Dargin ethnic group consists of three ethnicities: Dargins and two minor ethnicities—Kubachinians and Kaitagians.

We examined a mixed cohort of 191 non-related Dagestanians (i.e., students of various years from the Dagestan State University). One carrier of the c.3577del variant was detected. After that, we examined two national cohorts: Dargins, consisting of 132 non-related healthy residents of the republic, and Kubachinians, consisting of 46 people. Among the 132 Dargins, we found three c.3577del carriers, and among the Kubachinians no carriers were detected. Thus, the variant carrier frequency among Dargins was 2.3%, and the disease frequency among Dargins was 1.3 per 1000 people.

We analyzed five microsatellite markers: D3S3559, D3S3563, 3S1588, D3S1289, and D3S3666 from the region of the *COL7A1* gene; haplotype analysis was carried out on the material of eight non-related patients with the deletion in a homozygous state. The genotyping results are presented in Table 1. The presumed founder haplotype is highlighted in grey.

As a comparison group, we genotyped the DNA samples of 34 non-related Dargins who were without the c.3577del mutation, by using the same microsatellite markers. We determined the frequency of alleles in 16 carrier (D) chromosomes of homozygotes and 68 control chromosomes (N) for the 5 microsatellite loci positioned above (D3S3559 and D3S3563) and below (D3S1588, D3S1289, and D3S3666) the *COL7A1* gene on the Marshfield genetic map. The F-test results for the alleles of the markers with the highest linkage disequilibrium values δ are presented in Table 2.

Thus, the 4-3-7-4-9 haplotype for markers D3S3559, D3S3563, D3S1588, D3S1289, and D3S3666 is most likely the founder haplotype, which underwent gradual blurring, and the accumulation of the c.3577del mutation in the *COL7A1* gene in Dargins from the Republic of Dagestan is linked to the founder effect. The absence of the deletion in 46 Kubachinian Dagestan residents, who live in the same territory as Dargins and speak a language originating from the same linguistic family, confirms the distinct national assortativity of marriages within the territory of the republic.

The second most frequent mutations are mutations in the *KRT14* gene (NM_000526.5), which caused the development of the autosomal-dominant EB simplex in 28 children from 22 (13%) examined families (Figure 2). Eight mutations in the *KRT14* gene were described previously. Seven of these were missense mutations, and another one mutation interrupted the splice site. Four variants were novel: two deletions—c.1151_1180del, p.(L384_Q393del), and c.1231_1233del, p.(E411del); as well as two missense mutations—c.1234A>C, p.(Ile412Leu), and c.1235T>C, p.(I412T). Mutation c.356T>C, p.(M119T) was detected in four children with a severe generalized phenotype from four (1.7%) non-related families. Further, it was described earlier in patients with the same disease course [47]. The most common mutation in the *KRT14* gene, in our cohort, was the missense mutation c.373C>T, p.(R125C), which was detected in seven children (Figure 4).

We examined the origin of dominant mutations in the *KRT14* gene in 22 families: in 9 (40%) families, mutations were de novo, which is why healthy parents had an affected child; in 6 families, mutations were paternally inherited; in three families, mutations were maternally inherited; and in 4 cases, no parents’ biological material or family history were available.

The third most frequent mutations were the dominant mutations in the *KRT5* gene (NM_000424.4). These caused instances of EB simplex in 16 patients from 16 (6.5%) examined families (Figure 5). Among 14 various mutations, 4 variants have not been described previously in the HGMD database. These were three novel missense mutations c.355G>T, p.(G119C); c.592A>C, p.(T198P); and c.1340C>A, p.(A447D), as well as one stop-loss variant c.1772A>C, p.(*591Sext*37). Among ten other mutations (which were described earlier), eight were detected once, a missense mutation c.556G>C, p.V186L was detected twice, and a missense mutation c.545T>C, p.(F182S) was detected in three children from non-related families in the Kaliningrad region. This definitely shows an equal distribution of 14 out of 16 detected variants in the *KRT5* gene (Figure 5).

We examined the origin of the dominant mutations in the *KRT5* gene in 16 families: 5 mutations were de novo; 3 mutations were paternally inherited; 5 mutations were maternally inherited; and in 2 cases, no parental biological material or family history were available.

The molecular genetic testing performed in other countries showed that mutations in the *KRT5* and *KRT14* genes are responsible for the majority of EB simplex cases and are found in 70–75% of patients [18,48]. This finding clearly correlates with our data.

Mutations in the *COL17A1* gene (NM_000494.4) were the cause of junctional EB in 13 children from 13 (5%) families (Figure 6). Six out of seven detected mutations have not been described in the HGMD database previously. There were three novel missense mutations: homozygous c.1445T>C, p.(L482P) detected in one child; homozygous c.2488G>A, p.(G830R) detected in five children from the Chechen Republic; and three children from the Republic of Dagestan. In addition, there were also the missense mutation c.2725G>C, p.(G909R) detected in a compound heterozygous state with the c.3689dup duplication, which causes a p.(V1231Cfs*11) reading frame shift; one homozygous nonsense mutation c.3292C>T, which leads to a p.(Q1098*) premature translation termination in one child whose parents are cousins; and one homozygous splice site variant c.1268-2A>G (Figure 6).

The most frequent *COL17A1* mutation detected on 16 (62%) alleles in five families of the Chechen ethnic group and three families from Dagestan was c.2488G>A, which causes the p.(G830R) missense mutation. Two other homozygous variants of the *COL17A1* gene were detected in residents of Dagestan and Ingushetia. This can be seen as evidence of either a high rate of consanguineous marriages among North Caucasian ethnic groups, (having said that we are only well aware of one case), or recessive allele accumulation under the influence of genetic drift in small populations.

In six children with a severe general phenotype, we found mutations in the *LAMB3* gene (NM_000228.3). Among four detected novel mutations, there were two frame-shift duplications: c.1969_1970dup, p.(L658Pfs*50), c.2418dup, and p.(P807Sfs*31). Additionally, there were two splice site variants: c.3051+1G>T and c.3052-1G>A. However, we must mention that we did not detect mutations in the *LAMA3* and *LAMC2* genes within the examined children, which is notable as such mutations are most common in patients with junctional EB, according to foreign publications [49]. This fact could be a reflection of the significant inter-population differences.

Recently, peeling skin syndrome type 2 was added into the EB classification as a subtype of EB simplex [10]. We detected genetic variants of this syndrome in nine children from eight (3.2%) families. The mutations in the *TGM5* gene (NM_201631.4) were represented by six different variants: three novel small frame-shift deletions—c.331del, p.(A111Pfs*7), c.413_425del, p.(L138Pfs*21), and c.773del, p.(S258Tfs*5); as well as three missense mutations—a novel c.1036C>G, p.(R346G) and the (previously described) c.103C>T, p.Arg35Trp, and c.337G>T, p.(G113C). Further, the latter was detected in a homozygous state in four unrelated families.

We detected only novel variants among seven mutations in the *KRT10* gene (NM_000421.5) within children with ichthyosiform erythroderma from four (1.8%) families (Figure 2). All revealed variants were de novo mutations. All variants were identified once, except for the missense mutation c.467G>A, p.R156H that was revealed in two children from unrelated families.

We detected mutations in the *PLEC* gene (NM_201384.2) in five children from four (1.6%) families, which cause EB simplex. A total of six different variants were identified. All of them were novel: two nonsense mutations c.9508G>T and c.1555A>T, which lead to p.(E3170*) and p.(K519*) premature translation termination, respectively; two missense mutations c.979G>A, p.(G327S) and c.1108T>C, p.Trp370Arg; and one frameshift deletion c.4719_4723del, p.(S1573Rfs*37), which was found in the homozygous state in two siblings.

In three patients with Kindler syndrome from the Leningrad region and from Dagestan, we revealed three novel mutations in the *FERMT1* gene (NM_017671.5): two different deletions—c.778del, p.(Q260Kfs*21) and c.1088del, p.(L363Wfs*39) in a compound heterozygous state in one patient; a splicing mutation c.1139+1G>A in a homozygous state in another patient; and the third patient was a carrier of a previously described small deletion c.994_995del, p.(E332Gfs*9).

We found only one missense mutation described multiple times: c.556A>T, p.(N186Y) in the *KRT2* gene in a child with superficial epidermolytic ichthyosis.

For the dominant forms, we established either the inheritance of pathogenic variants from one of affected parents: paternal (17), maternal (18), or de novo mutations (29) in cases when both biological parents have no clinical signs of EB. For recessive forms, the segregation of new pathogenic variants from heterozygous parents was also established.

In total, we detected 182 causative variants in 10 out of 33 studied genes. Less than a half, 89 (46%), were missense mutations; 34 (18%) were splice site mutations; 29 (15%) were small deletion mutations; 23 (12%) were nonsense mutations; 14 (7%) were small insertion mutations; 2 (1%) were regulatory mutations; and 1 (1%) was a stop codon mutation. We detected LoF variants (mutations that reduce the length of the encoded protein, i.e., nonsense and reading frame shift mutations) in 68 (25%) patients. This type of mutation correlates with the severity of clinical signs, especially in cases of homozygous variants.

In total, we detected 103 novel mutations, which is 56% of all the detected mutations. Further, this may enlarge the HGMD database by 4.1% in the examined genes. The high rate of non-described mutations is showing the high heterogeneity and insufficient knowledge of genetic causes of hereditary dermatosis in patients from the Russian Federation. The high rates of novel mutations stand out in comparison with other studies on EB genetic causes, as well [41,43].

## 3. Materials and Methods

A total of 268 Russian children from 247 families—131 boys and 137 girls aged from 1 month to 17 years and 10 months—with clinical manifestations of the epidermolysis bullosa symptom complex were selected for the purposes of conducting molecular genetic diagnostics.

### 3.1. Molecular Diagnostics

Target regions of 33 genes were analyzed using next generation sequencing (NGS). Enrichment was carried out using SeqCapEZ technology (Roche, Santa Clara, CA, USA). The total size of the panel was 104,950 bp, the average reading depth was 240×, and the number of reads with 50× depth was more than 99% in all target areas. A semiconductor Ion S5 sequencer PCR (Thermo Fisher Scientific, Waltham, MA, USA), Miseq (Illumina, San Diego, CA, USA) and Nextseq500DX (Illumina, USA, San Diego) were used as the sequencing platforms.

All detected genome missense variants with a frequency less than 1%—according to the international base gnomAD, version 2.1.1 (http://gnomad.broadinstitute.org, accessed on 16 November 2022)—and absent from the HGMD prof database (version 2020.1. June 2020, https://portal.biobase-international.com/hgmd/pro accessed on 16 November 2022) underwent bioinformatic analysis. As a result, we separated the variants that possessed pathogenicity as confirmed by at least three out of four bioinformatic resources: SIFT (damaging), PolyPhen-1 (probably damaging), PolyPhen-2 (probably damaging), and Mutation Tester (disease causing). The conserved novel missense and splicing mutations were analyzed using the bioinformatic software, Alamut Visual Plus (version 1.5.1 SOPHiA GENETICS, Lausanne, Switzerland).

The reference nucleotide sequences were chosen from the NCBI database (National Center for Biotechnological Information, Bethesda, Rockville, MA, USA). The BLAST program (https://blast.ncbi.nlm.nih.gov/Blast.cgi, accessed on 16 November 2022) was used in order to determine conservative regions. The oligonucleotide sequences were selected using the program Beacon Designer 8.10. The specificity of the primer pairs was checked with Primer-BLAST (https://www.ncbi.nlm.nih.gov/tools/primer-blast, accessed on 16 November 2022). For Sanger sequencing, the coding and flanking intronic regions (20 bp) of the targeted genes were amplified using a ProFlex™ PCR System (Thermo Fisher Scientific, USA, Waltham). Amplification products were sequenced using a BigDye^®^ Terminator v3.1 Cycle Sequencing Kit (Thermo Fisher Scientific, USA, Waltham) on an ABI 3500xl Genetic Analyzer (Thermo Fisher Scientific, USA, Waltham). Data were analyzed using the ABI Data Collection software v3.0 (Thermo Fisher Scientific, USA, Waltham) and Sequencing Analysis software 5.2 (Thermo Fisher Scientific, USA, Waltham). The sequences were compared to the reference DNA sequence (GenBank Accession https://www.ncbi.nlm.nih.gov/genbank/, accessed on 16 November 2022). Segregation analysis allowed us to confirm the pathogenicity of all novel variants.

The c.3577del mutation in the *COL7A1* gene was detected using a custom system based on allele-specific ligase reaction. The products were separated via electrophoresis in a polyacrylamide gel with subsequent ethidium bromide staining and UV visualization.

We examined the following five microsatellite markers from 17p13.2 (2.44-kb region around the *COL7A1* gene): D3S3559, D3S3563, DS1588, D3S1289, and D3S3666. All markers were chosen using the Marshfield NCBI genetic map. The microsatellite markers were examined via AFLP analysis. The DNA fragments were amplified using PCR. The results were registered via electrophoresis in a polyacrylamide gel with subsequent ethidium bromide staining and UV visualization.

### 3.2. Statistical Analysis

The statistical analysis of the allele frequencies on the mutant chromosomes and the control group chromosomes was based on the χ^2^ test for a 2 × 2 contingency table that showed the comparison of the two groups: associated allele and all other alleles. To evaluate linkage disequilibrium (LD), we used the following formula: δ = (PD − PN)/(1 − PN), where PD is the frequency of the associated allele on mutant chromosomes and PN is the frequency of the same allele on the normal chromosomes (Bengtsson and Thomson, 1981).

## 4. Conclusions

The custom diagnostic method of hereditary dermatosis with the EB symptom complex using NGS has allowed us, for the first time in Russia, to conduct molecular genetic testing in 268 children from 247 families, as well as to detect 192 various pathogenic variants in 11 out of 33 examined genes. An analysis of all the obtained data allowed us to estimate the frequencies of the different EB types in the affected Russian children. We have characterized not only the population and geographical features (as well as primary encounter rates of certain mutations specific to Russian children), but also the genetic traits of North Caucasus ethnic groups. Thus, the most frequent mutation in the examined children—c.425A>G, p.(K142R) in the *COL7A1* gene; the deletion c.3577del, p.(A1193Lfs*72) in the *COL7A1* gene; and the missense mutation c.2488G>A, p.(G830R) in the *COL17A1* gene—are the most typical mutations for Dargwa and Chechen ethnic groups. The study of the haplotypes of microsatellite markers that we were able to perform in the Dargin population, confirmed the presence of the founder effect.

We have revealed the number of mutations in the *COL17A1* gene in Chechens. This opens up a new area for further research, which is the determining of the founder effect in the Chechen population via microsatellite markers.

The current research allowed us to detect 103 novel pathogenic genetic variants, which are not present in the HGMD database.

Thus, NGS can be used for the purposes of the identification of the genetic causes of hereditary dermatosis with the EB symptom complex. This, along with comprehensive clinical examination, allows one to make the correct diagnosis as early and rapidly as possible, avoiding the invasive procedure of skin biopsy and the inconclusive results of IF staining. Therefore, eliminating the need for performing Sanger sequencing of several genes.

## Figures and Tables

**Figure 1 ijms-23-14343-f001:**
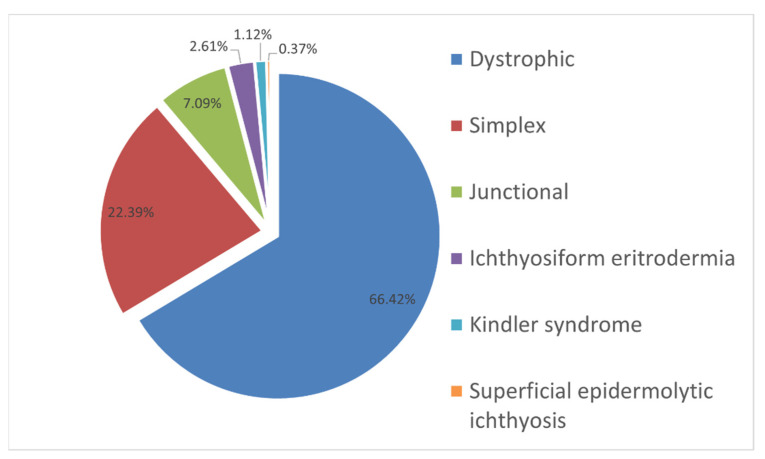
The incidence expressed as percentage cases of different dermatosis forms with the EB symptom complex in examined Russian children.

**Figure 2 ijms-23-14343-f002:**
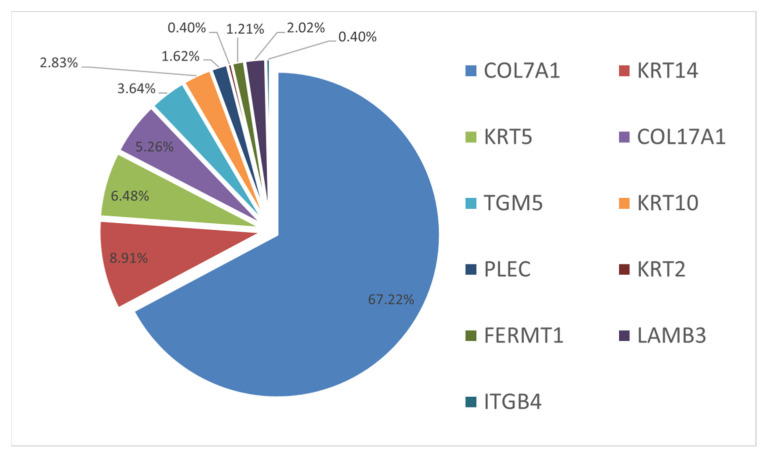
Variety of genes and the proportion of mutations in 247 examined families with the EB phenotype.

**Figure 3 ijms-23-14343-f003:**
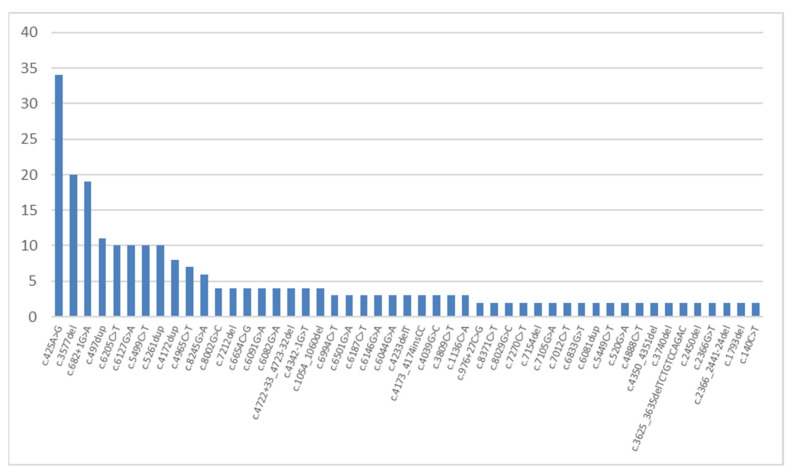
The frequencies and spectrum of *COL7A1* gene mutations detected more than twice in Russian children with dystrophic EB.

**Figure 4 ijms-23-14343-f004:**
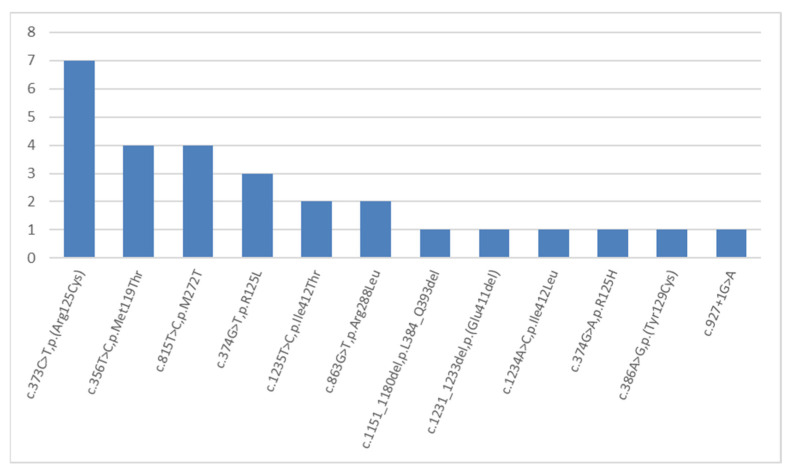
The frequencies and spectrum of detected mutations in the *KRT14* gene in Russian children with an EB phenotype.

**Figure 5 ijms-23-14343-f005:**
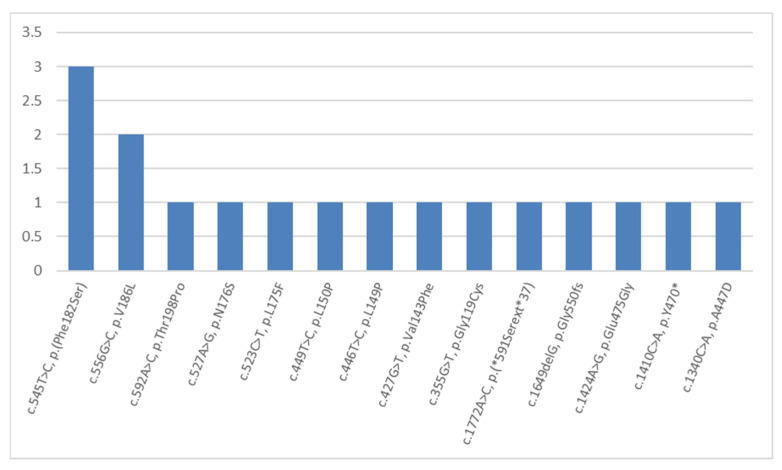
The frequencies and spectrum of detected mutations in the *KRT5* gene in Russian children with EB simplex.

**Figure 6 ijms-23-14343-f006:**
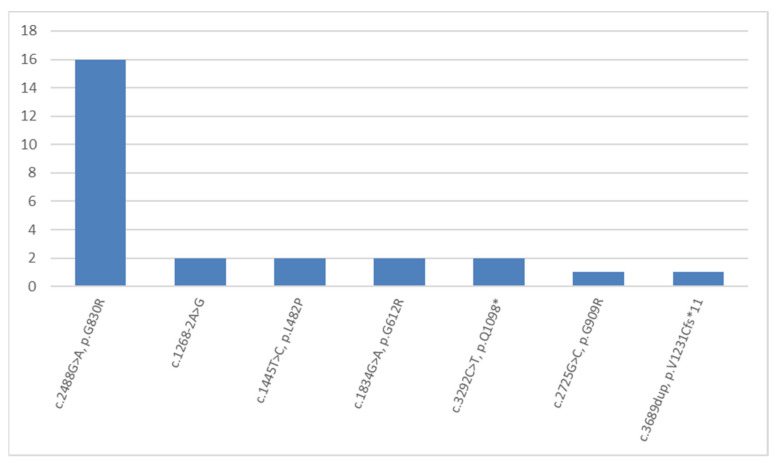
The frequencies and spectrum of detected mutations in the *COL17A1* gene in Russian children with junctional EB.

**Table 1 ijms-23-14343-t001:** Haplotypes of chromosomes with the c.3577del mutations using markers D3S3559, D3S3563, D3S1588, D3S1289, and D3S3666. The assumed ancestral haplotype is highlighted in gray.

Marker		D3S3559	D3S3563	*COL7A1*	D3S1588	D3S1289	D3S3666
	Coordinates (cM)	67.49	68.47	70.61	70.61	71.41	72.21
Chromosome							
9.1		4	3	c.3577del	7	4	9
8.1		4	3	c.3577del	7	4	9
8.2		1	3	c.3577del	7	4	7
6.1		5	4	c.3577del	7	4	9
6.2		4	1	c.3577del	7	4	8
9.2		4	3	c.3577del	6	0	5
1.1		4	3	c.3577del	6	4	9
1.2		4	3	c.3577del	6	4	8
4.1		4	3	c.3577del	6	4	9
2.1		4	3	c.3577del	1	4	9
3.1		4	3	c.3577del	1	4	8
5.1		6	3	c.3577del	7	5	4
5.2		6	3	c.3577del	7	5	4
2.2.		2	3	c.3577del	1	4	8
3.2		2	3	c.3577del	1	4	7
4.2		4	6	c.3577del	7	8	8

**Table 2 ijms-23-14343-t002:** Linkage disequilibrium analysis between *COL7A1* c.3577del mutation and microsatellite markers close to the *COL7A1* gene.

Marker	Coordinates, cM	Allele	*p*-Value	δ ± 95 CI
D3S3559	67.49	4	*p* < 0.05	0.61 ± 0.13
D3S3563	68.47	3	*p* < 0.05	0.77 ± 0.12
D3S1588	70.61	7	*p* < 0.05	0.44 ± 0.15
D3S1289	71.41	4	*p* < 0.05	0.67 ± 0.15
D3S3666	72.21	9	*p* < 0.05	0.30 ± 0.15

## Data Availability

Data available on request due to privacy restrictions.

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
