# Peer review of "Targeted NGS in Diagnostics of Genodermatosis Characterized by the Epidermolysis Bullosa Symptom Complex in 268 Russian Children"

_ijms, 2022, doi:10.3390/ijms232214343_

Round 1
Reviewer 1 Report
In this interesting manuscript the authors have developed a diagnostic panel consisting of coding and intronic areas of 33 genes at the basis of EB. NGS sequencing has identified 192 various altered alleles of which 109 alleles not described so far. In addition, the authors have characterized incidences of different epidermolysis bullosa types.
Major point:
At the end of page 3 the authors mentioned that rare and novel genetic variants were detected in some examined genes “…but molecular genetic testing of parents did not allow us to assume these variants to be causative”. In order, to improve the manuscript value and to be useful for other research on this topic, the authors should report these data as supplementary file. They can report the data also considering the privacy restrictions.
Minor point
Since the incidence is a measure of the number of new cases. in the title of figure 1 could be better to write “The incidence expressed as percentage of cases…”
Author Response
Thank you for your response.
We agree that additional information about non-causal variants in our study will improve the manuscript value. That is why we add Appendix A, where we represented table with this information. This table also available in attach file.
We also rephrased the name of the first figure: "The incidence expressed as percentage cases of different dermatosis forms with the EB symptom complex in examined Russian children."
All changes can be found in resubmitted manuscript.

Reviewer 2 Report
The authors report on molecular genetic data in a large cohort of children with genodermatoses from Russia. Altogether the study provides significant infromation on epidemiology and genetic of these disorders in populations living in Russia.
The authors should adapt the terminology to the latest classifications of EB and ichthyoses.
The statement that this is the first such study in children is probably overstated, Instead focus should be on genetic data from populations living in Russia and on population specific mutations.
Author Response
Thank you for your response.
We renamed ichthyosis bullosa of Siemens into Superficial epidermolytic ichthyosis and pachyonychia congenital into pachyonychia congenita, according to the last classifications of EB and ichthyoses.
We rephrased, that our research was unique for Russian population.
All changes can be found in resubmitted manuscript.